# The Antimicrobial Resistance Characteristics of Imipenem-Non-Susceptible, Imipenemase-6-Producing *Escherichia coli*

**DOI:** 10.3390/antibiotics11010032

**Published:** 2021-12-28

**Authors:** Reo Onishi, Katsumi Shigemura, Kayo Osawa, Young-Min Yang, Koki Maeda, Shiuh-Bin Fang, Shian-Ying Sung, Kenichiro Onuma, Atsushi Uda, Takayuki Miyara, Masato Fujisawa

**Affiliations:** 1Department of Public Health, Division of Infectious Diseases, Kobe University Graduate School of Health Sciences, 7-10-2 Tomogaoka Suma-ku, Kobe 654-0142, Japan; ohnishile@gmail.com; 2Division of Urology, Kobe University Graduate School of Medicine, 7-5-2 Kusunoki-cho, Chuo-ku, Kobe 650-0017, Japan; yym1112@gmail.com (Y.-M.Y.); kokimaeda1118@gmail.com (K.M.); masato@med.kobe-u.ac.jp (M.F.); 3Department of Medical Technology, Kobe Tokiwa University, 2-6-2 Otani-cho, Nagata-ku, Kobe 653-0838, Japan; k-ohsawa@kobe-tokiwa.ac.jp; 4Department of Pediatrics, Division of Pediatric Gastroenterology and Hepatology, Shuang Ho Hospital, Taipei Medical University, 291 Jhong Jheng Road, Jhong Ho District, New Taipei City 23561, Taiwan; sbfang@tmu.edu.tw; 5Department of Pediatrics, School of Medicine, College of Medicine, Taipei Medical University, Taipei 11031, Taiwan; 6International Ph.D. Program for Translational Science, College of Medical Science and Technology, Taipei Medical University, Taipei 11031, Taiwan; ssung@tmu.edu.tw; 7Department of Infection Control and Prevention, Kobe University Hospital, 7-5-2 Kusunoki-cho, Chuo-ku, Kobe 650-0017, Japan; onumak@med.kobe-u.ac.jp (K.O.); a-uda@umin.ac.jp (A.U.); miyarat@med.kobe-u.ac.jp (T.M.)

**Keywords:** IMP-6, carbapenemase, porin, efflux pump, plasmid

## Abstract

Imipenemase-6 (IMP-6) type carbapenemase-producing *Enterobacteriaceae* is regarded as dangerous due to its unique lack of antimicrobial susceptibility. It is resistant to meropenem (MEPM) but susceptible to imipenem (IPM). In addition to carbapenemase, outer membrane porins and efflux pumps also play roles in carbapenem resistance by reducing the antimicrobial concentration inside cells. Extended-spectrum β-lactamase (ESBL) is transmitted with IMP-6 by the plasmid and broadens the spectrum of antimicrobial resistance. We collected 42 strains of IMP-6-producing *Escherichia coli* and conducted a molecular analysis of carbapenemase, ESBL, porin, efflux, and epidemiological characteristics using plasmid replicon typing. Among the 42 isolates, 21 strains were susceptible to IPM (50.0%) and 1 (2.4%) to MEPM. Seventeen strains (40.5%) co-produced CTX-M-2 type ESBL. We found that the relative expression of *ompC* and *ompF* significantly correlated with the MIC of IPM (*p* = 0.01 and *p* = 0.03, respectively). Sixty-eight% of CTX-M-2-non-producing strains had IncI1, which was significantly different from CTX-M-2-producing strains (*p* < 0.001). In conclusion, 50.0% of our IMP-6-producing strains were non-susceptible to IPM, which is different from the typical pattern and can be attributed to decreased porin expression. Further studies investigating other types of carbapenemase are warranted.

## 1. Introduction

The prevalence of imipenemase-6 (IMP-6) type carbapenemase-producing *Enterobacteriaceae* (CPE) is increasing in Japan, and this is regarded as dangerous due to its unique lack of antimicrobial susceptibility [1,2]. They are generally resistant to meropenem (MEPM), although they are susceptible to imipenem (IPM) [2,3]. Carbapenemase is the enzyme that inactivates carbapenem. CPE can produce a variety of carbapenemases, which are divided into three groups as class A, class B, and class D according to the Ambler classification [4]. In Japan, the most common carbapenemase type is IMP, one of the class B carbapenemases. It exhibits metallo-β-lactamase and is able to decompose almost all β-lactam drugs, including carbapenems, by a wide range of instrumental specificity. IMP-6 exceptionally shows characteristic inactivation as described above [5]. CPE often co-produces extended-spectrum β-lactamase (ESBL), which inactivates penicillins, cephalosporins, and monobactams [3,6] for wider spectrum antimicrobial resistance. IMP-6 generally does not inactivate penicillins and monobactams [2,7], but if an IMP-6-producing strain co-produces ESBLs, it does inactivate these antimicrobials. Thus, the co-production of carbapenemase and ESBL should be considered as an important mechanism in antimicrobial resistance.

Outer membrane porins and efflux pumps are also important mechanisms of carbapenem resistance *Enterobacteriaceae* (CRE) by reducing the antimicrobial concentration inside cells [8]. Because carbapenem is one of the most effective wide-spectrum antimicrobials, CRE infections increase treatment difficulty and worsen prognosis [9].

Porins are small pores in the outer membrane of Gram-negative bacteria (GNB). Decrease or loss of porins hinders drug entry in GNB [10]. Carbapenems enter cells through porins such as OmpC and OmpF, encoded by porin genes *ompC* and *ompF* [11]. OmpC and OmpF are the two most important outer membrane porin proteins in *Escherichia coli*. The pore size of OmpF is larger than that of OmpC, allowing more solutes, including noxious agents, to diffuse into the cell through the OmpF channel [12]. A decrease in the relative expression of *ompC* and *ompF* indicates loss or disruption of OmpC and OmpF.

Efflux pumps are responsible for membrane transport and can discharge antibacterial drugs outside the cell. Hyperfunction of efflux pump proteins is one of the most important mechanisms in carbapenem resistance. Resistance-nodulation-division (RND) efflux pumps are a major mechanism of multidrug resistance in *Enterobacteriaceae* [13]. The AcrAB-TolC RND system is the most common among the different efflux systems, which catalyze substrate efflux by an H^+^ antiport mechanism [13,14]. Phenylalanine-arginine β-naphthylamide (PAβN, also called MC-207,110) is well known as a broad-spectrum efflux pump inhibitor [15]. PAβN binds to AcrB, leading to the inhibition of substrate efflux outside the cell [16,17]. In carbapenem resistance attributed to the overexpression of efflux pumps, the minimum inhibitory concentration (MIC) will decrease from the addition of PAβN because carbapenem cannot be effluxed out of the cell. Therefore, efflux pump activity in carbapenem resistance can be evaluated using PAβN [18,19].

Plasmids are extra-chromosomal circular fragments of DNA that replicate autonomously in host cells [20]. Plasmids are classified based on incompatibility (Inc) groups by Plasmid Replicon Typing [20]. Carbapenemase genes and ESBL genes are transferred easily by plasmids, contributing to the rapid spread of carbapenemase and ESBL-producing strains. IncN plasmid often carries IMP-6 with CTX-M-2 type ESBL and plays an important role in the spread of IMP-6 and CTX-M-2 co-producing strains [3,20]. Plasmid replicon typing facilitates the exploration of the epidemiological features of carbapenemase-producing strains.

Whether resistance mechanisms other than carbapenemase affect antimicrobial susceptibility in IMP-6-producing bacteria has rarely been investigated. An epidemiological analysis is also important for a comprehensive understanding of the spread of CPE. The purpose of this study was to assess antimicrobial susceptibility and the effects of the three carbapenem resistance mechanisms, ESBL production, and epidemiology in IMP-6-producing *Escherichia coli*.

## 2. Results

### 2.1. Antimicrobial Susceptibilities

Among the 42 IMP-6-producing isolates, 21 strains (50.0%) were susceptible to IPM, 1 strain (2.4%) to MEPM, no strains (0%) to ertapenem (ETP), and 11 strains (26.2%) to doripenem (DRPM). Minimum inhibitory concentrations (MIC)_50_ were 1.5 μg/mL, 16 μg/mL, 16 μg/mL, and 8 μg/mL, and MIC_90_ were 4 μg/mL, 32 μg/mL, 64 μg/mL, and 32 μg/mL for IPM, MEPM, ETP, and DRPM, respectively (Table 1) (Appendix A). Most strains were resistant to piperacillin (PIPC) (97.6%), ceftazidime (CAZ) (97.6%), cefepime (CFPM) (95.2%), ciprofloxacin (CPFX) (97.6%), and levofloxacin (LVFX) (97.6%). No strain was resistant to amikacin (AMK) (0%).

### 2.2. Phenotypic and Genotypic Detection of Carbapenemase and ESBL

Carbapenemase-associated genes other than *bla*_IMP-6_ were not detected in our strains. Twenty-nine strains (69.0%) had confirmed ESBL production and ESBL *bla*_CTX-M-2_ genes were detected in 17 strains (40.5%), *bla*_CTX-M-14_ in 1 strain (2.4%), and *bla*_CTX-M-15_ in 2 strains (4.8%) (Table 2) (Appendix A).

### 2.3. Relative Expression Level of ompC and ompF

Relative expression levels of *ompC* and *ompF* were measured by quantitative reverse transcription polymerase chain reaction (qRT-PCR). The relative expression levels of *ompC* significantly correlated with the MICs of IPM (r = −0.388, *p* = 0.0112) and DRPM (r = −0.501, *p* = 0.000734) (Figure 1A,D). Furthermore, the relative expression levels of *ompF* significantly correlated with the MICs of IPM (r = −0.332, *p* = 0.0318) and MEPM (r = −0.529, *p* = 0.00032) (Figure 2A,B).

### 2.4. Efflux Pump Activity and Carbapenem Resistance

To determine the effect of efflux on IPM resistance, the MICs of two representative carbapenems, IPM and MEPM, were measured with and without the addition of PAβN (Figure 3). If carbapenem resistance depends on the overexpression of efflux pumps, MICs of carbapenems will decrease with the addition of PAβN because carbapenem cannot be effluxed out of the cell. In this study, no isolates showed more than a four-fold decrease of MIC in the presence of PAβN compared with MICs when only IPM and MEPM were present. Therefore, no significant effect of efflux pump activity on carbapenem resistance was found in all 42 strains. It was confirmed by the control that PAβN did not inhibit bacterial growth (data not shown).

### 2.5. Plasmid Replicon Typing

IncF was the most prevalent plasmid replicon, identified in 40 of the 42 strains (95.2%) (Table 3). A variety of plasmids were found in our IMP-6-producing *E. coli* strains, including IncFIA in 37 strains (88.1%), IncN in 36 strains (85.7%), IncFIB in 28 strains (66.7%), IncI1 in 18 strains (42.9%), IncB/O in 4 strains (9.5%), and IncA/C in 2 strains (4.8%). There was a significant difference in the rates of carrying IncI1 between CTX-M-2 co-producing strains (*n* = 17) and the others (*n* = 25) (*p* < 0.001, Table 3). We found duplications of plasmids in the isolates. The details for individual strains are presented in Appendix A.

## 3. Discussion

IMP-6-producing CRE is frequently detected in Japan, especially in west Japan. A previous study from Osaka showed that 130 (97.0%) of 134 carbapenem-resistant *E. coli* produced IMP-6 [1]. Their susceptibility rates to MEPM and IPM were 0% and 100.0%, respectively, which is a typical pattern of IMP-6-producing CRE. Our IMP-6-producing *E. coli* strains were resistant to at least one of the carbapenems, and most of these strains were resistant to MEPM (92.9%). Although most previous studies indicated that IMP-6 does not inactivate IPM [2,3], 50.0% of our strains were non-susceptible to IPM. Thus, non-susceptibility to IPM may be due to a resistance mechanism other than the production of IMP-6. A few studies reported the susceptibility of IMP-6-producing *Enterobacteriaceae* to ETP and DRPM. A previous study in Korea reported that all of 10 IMP-6-producing strains were not susceptible to ETP [21], similar to our strains. Another study in Japan showed that all of 11 IMP-6-producing strains were not susceptible to DRPM [22], which is consistent with the low susceptibility (26.2%) in our strains. Altogether, these results demonstrated the non-susceptibility of IMP-6-producing strains to ETP and DRPM.

Previous studies reported that most IMP-6-producing strains co-produced CTX-M-2 [7,23], but less than half of our strains co-produced CTX-M-2. The form of transmission may have changed. Furthermore, most of our strains were resistant to PIPC with or without CTX-M-2 production, despite the fact that IMP-6-producing strain is generally susceptible to PIPC. Quite possibly, these strains co-produced another β-lactamase. For example, a previous study reported that IMP-6-producing strains co-produced TEM-1, a non-ESBL β-lactamase [2]. These strains were resistant to PIPC by co-producing TEM-1. 

One of the main mechanisms for carbapenem resistance other than carbapenemase production is the decrease or loss of porin. A previous study in China showed low expression of either *ompC* or *ompF* in 46 (68.7%) of 67 carbapenemase-non-producing strains [24]. In that study, porin-related mechanisms were investigated only for carbapenemase-non-producing strains, not for carbapenemase-producing strains. Our study explored porin-related mechanisms in CPE and found that IMP-6-producing CRE had decrease or loss of porin according to the downregulation of two porin genes *ompC* and *ompF* as another resistance mechanism. Furthermore, statistical analysis showed a significant negative correlation between the expression levels of the two porin genes and MIC of some carbapenems, including *ompC* with IPM and DRPM (Figure 1A,D) as well as *ompF* with IPM and MEPM (Figure 2A,B). Although half of the IMP-6-producing CRE strains in our study were susceptible to IPM, we still found that the MICs of IPM were correlatively higher when the porin-related genes *ompC* and *ompF* were more downregulated. Thus, decreased porin expression can also affect antimicrobial resistance in IMP-6-producing strains, with altered antimicrobial susceptibility in some cases. 

Efflux activity in carbapenem resistance has been reported using several methods for confirmation of efflux activity [19,24]. A previous study showed that 7 isolates of carbapenem-resistant *Pseudomonas aeruginosa* exhibited a significant reduction in meropenem MIC with PAβN, using the same methods as our study [25]. Similar to our study, another group investigated KPC-2-producing CRE and did not find any effect of efflux pump activity in carbapenem resistance [19]. Our strains also did not show a contribution by efflux pump activity to carbapenem resistance. Taken together, these results suggest the minor effect of efflux activity on carbapenem resistance in carbapenemase-producing bacteria. 

Plasmid replicon typing is a useful assay to investigate horizontal gene transfer because carbapenemase genes and ESBL genes are generally encoded on plasmids [3,20]. As mentioned above, IMP-6-producing CRE strains often possess IncN plasmid and simultaneously produce CTX-M-2 [3]. A previous study showed that 80 strains (96.4%) possessed IncN, and 73 strains (88.0%) co-produced CTX-M-2 among 83 IMP-6-producing strains [7]. In contrast, a lower portion (40.5%, 17/42) of our strains co-produced IMP-6 and CTX-M-2 despite a high carriage rate (85.7%, 36/42) of IncN. Compared with CTX-M-2 co-producing strains, IMP-6 producing but not CTX-M-2 co-producing strains significantly possessed IncI1 (*p* < 0.001). So far, IncI1 has not been reported in IMP-6-producing CRE. Transmission and carriage of IncI1 type plasmids could reduce co-production of CTX-M-2 among IMP-6-producing CRE.

Our study confirmed that IMP-6-producing strains were resistant to MEPM, ETP, and DRPM as reported so far. However, the IPM non-susceptible strains appeared to differ from the typical pattern. We demonstrated that downregulation of porin-associated genes was responsible for this IPM non-susceptibility, suggesting evolving characteristics of IMP-6-producing CPE. It is necessary to update our knowledge of the antimicrobial susceptibility of IMP-6-producing CPE to facilitate appropriate treatment and clinical practice. Downregulated porin-related genes should also be taken into account when developing antimicrobials against IMP-6-producing CPE.

There are some limitations in this study, including the small number of isolates and the unavailability of full clinical data for investigation. In addition, phenotypic studies of porin and genotypic efflux pump analysis were not performed. Despite these limitations, our study contributes to preventing the further expansion of IMP-6-producing CRE.

## 4. Materials and Methods

### 4.1. Bacterial Collection

Forty-two strains of IMP-6-producing *E. coli* were collected from Hyogo prefecture and sent to Hyogo Clinical Laboratory Corporation, Himeji, Japan, from 2012 to 2018. They were isolated from urine, sputum, and other sources. IMP-6 production was confirmed using phenotypic and genotypic methods. Carbapenemase production was detected by three phenotypic methods, the carbapenem inactivation method (CIM), double disk synergy test (DDST) with sodium mercaptoacetic acid (SMA), and the modified Hodge test. We defined a strain as carbapenemase-producing if it was positive by at least one method. CIM and modified Hodge tests were carried out according to Clinical and Laboratory Standards Institute (CLSI) recommendations [26]. SMA-DDST was performed for CAZ and IPM as previously described [27]. We detected *bla*_IMP-6_ by PCR and DNA sequencing as previously described [28,29].

### 4.2. Antimicrobial Susceptibility Tests

Antimicrobial susceptibility testing was performed by broth microdilution methods according to the CLSI guidelines [26]. MIC of *E. coli* was tested for the following antimicrobials: MEPM (Wako, Fujifilm Wako Pure Chemical Corporation, Japan), IPM (Wako, Japan), ETP (Sigma-Aldrich, St. Louis, MO, USA), and DRPM (Wako, Japan), using *E. coli* ATCC 25922 as quality control. MicroScan (Beckman Coulter Inc., Brea, CA, USA) was used to determine the MIC of the antimicrobials PIPC, CAZ, CFPM, LVFX, CPFX, AMK, and GM. We determined susceptibility according to the current CLSI clinical breakpoint [26]. We defined non-susceptible as resistant or intermediate to each antimicrobial.

### 4.3. Phenotypic Detections of ESBL Production in E. coli Isolates

All isolates of *E. coli* were screened for ESBL production using cefotaxime (CTX) and CAZ disc diffusion testing, according to CLSI guidelines [26]. The confirmation tests were conducted by the double-disk synergy method using CTX and CAZ disks alone and in combination with clavulanic acid [26].

### 4.4. Detection of Carbapenemase Genes and ESBL Genes

The presence of carbapenemase genes *bla*_IMP-1_, *bla*_IMP-2_, *bla*_KPC_, *bla*_NDM_, *bla*_OXA-48_ *bla*_GES_, *bla*_VIM-1_, and *bla*_VIM-2_ was determined using PCR amplification, as previously described [28,30,31,32,33,34]. The presence of ESBL genes *bla*_CTX-M_, *bla*_TEM_, and *bla*_SHV_ were also determined using PCR amplification, as previously described [35]. The PCR products were run on 1% agarose gel and stained with ethidium bromide (0.5 mg/mL) in a dark room. PCR purification was conducted by QIAquick PCR Purification Kit (Qiagen, Hilden, Germany) with sequencing by Eurofins Genomics, Inc. (Tokyo, Japan) [29].

### 4.5. q-RT-PCR for Porin Coding Genes ompC and ompF

The qRT-PCR for porin genes (*ompC* and *ompF*) was performed. RNA was extracted from bacterial pellets and DNase-treated using NucleoSpin RNA (Macherey-Nagel, Germany). cDNA was constructed using the ReverTra Ace qPCR RT Kit (Toyobo, Japan). The qRT-PCR was performed for porin genes *ompC* and *ompF* and the housekeeping gene *rpoB* using the primer previously described [11]. The reactions were run on a CFX Connect (Bio-Rad, Hercules, CA, USA) with the following cycling parameters: 95 °C for 5 min and 40 cycles of 95 °C for 15 s, 60 °C for 30 s, and 72 °C for 30 s, followed by melting with ramping from 60 °C to 95 °C in 0.2 °C increments. Melting curve analysis was performed to identify the amplicons. Each experiment was performed in triplicate, and results were presented as the mean value of three experiments. The relative expression levels were calculated by the 2^−ΔΔCt^ method. Expression of porin genes was normalized using the housekeeping gene *rpoB* in the same sample. Fold change in porin expression was determined by calculating the ratio of normalized porin expression in IMP-6-producing isolates to the control strain of the same species, *E. coli* ATCC 25922 [11]. 

### 4.6. Efflux Pump Inhibitory Assay

To examine the influence of efflux pumps on carbapenem resistance, efflux inhibitory assays were performed using representative carbapenem, IPM, and MEPM. The MICs of imipenem and meropenem were measured with and without 25 mg/L of the efflux pump inhibitor PAβN (Sigma-Aldrich, St Louis, MI, USA) [36]. Efflux pump activity was evaluated by fold change from the MIC without PAβN to the MIC with PAβN. If carbapenem resistance depended on the overexpression of efflux pumps, the MIC of carbapenems would be reduced by the addition of PAβN because carbapenem does not efflux. More than a four-fold decrease of the MICs in the presence of PAβN compared with the MICs in the absence of PAβN was defined as positive.

### 4.7. Plasmid Replicon Typing

Plasmid DNA was extracted as previously described [29]. Plasmid replicon typing was performed to see the disseminating formulation and investigate for IncF (FIA, FIB, FIC, F, FII), H, I, L/M, N, P, W, T, A/C, K, B/O, X, and Y by PCR amplification. The temperature conditions were initial denaturing at 94 °C for 5 min, followed by 30 cycles of denaturation at 94 °C for 1 min, annealing at 60 °C for 30 s, extension at 72 °C for 1 min and final extension at 72 °C for 5 min [20].

### 4.8. Statistical Analysis

Correlations between the relative expression of porin genes (*ompC* and *ompF*) and the MIC for each carbapenem were analyzed by Spearman’s rank correlation coefficient using EZR (Saitama Medical Centre, Jichi Medical University, Saitama, Japan) [37]. The difference in plasmid carriage between CTX-M-2-producing strains and the others was analyzed by Fisher’s exact test using EZR. Statistical differences among mean values were considered significant when *p* < 0.05.

## 5. Conclusions

Half of our IMP-6-producing strains were non-susceptible to IPM, and only 40.5% of them co-produced CTX-M-2, unlike the well-known typical pattern. Downregulation of porin-associated genes is responsible for this alteration in IPM non-susceptibility, but efflux plays a minor role. Further studies are warranted for investigation in other types of carbapenemase.

## Figures and Tables

**Figure 1 antibiotics-11-00032-f001:**
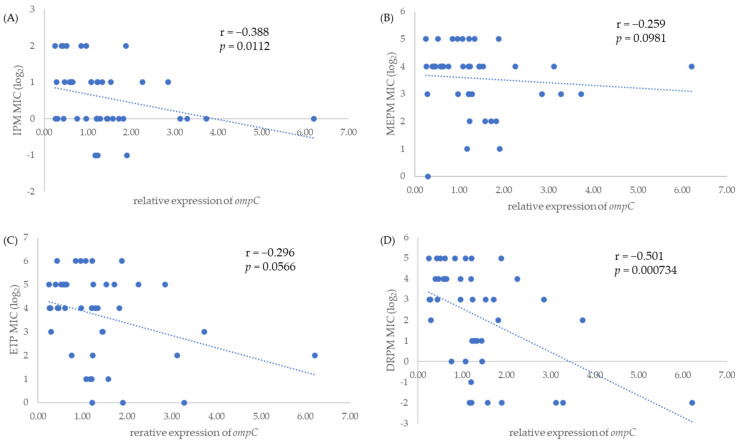
Correlation between relative expression levels of *ompC* and MICs of four carbapenems, IPM (**A**), MEPM (**B**), ETP (**C**), and DRPM (**D**), among IMP-6-producing *E. coli* strains.

**Figure 2 antibiotics-11-00032-f002:**
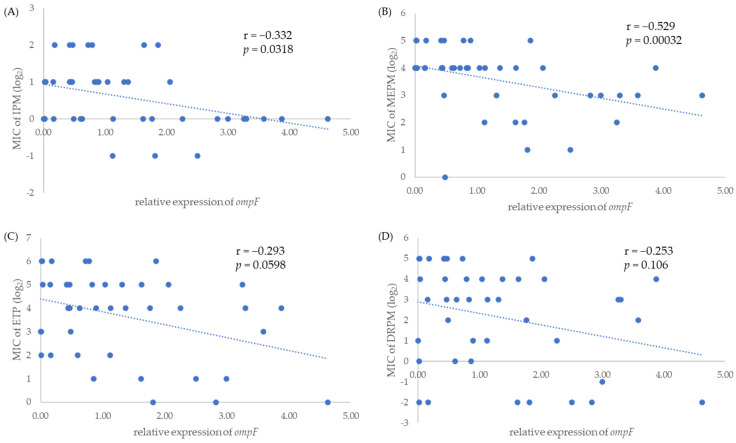
Correlation between relative expression levels of *ompF* and MICs of four carbapenems, IPM (**A**), MEPM (**B**), ETP (**C**), and DRPM (**D**), among IMP-6-producing *E. coli* strains.

**Figure 3 antibiotics-11-00032-f003:**
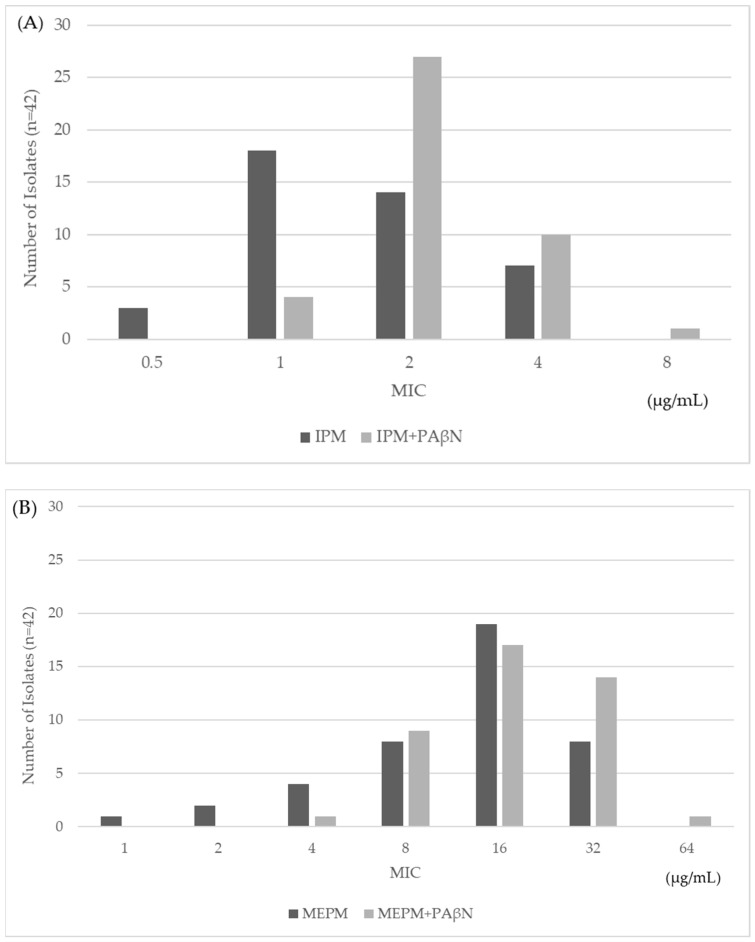
The MICs of IPM (**A**) and MEPM (**B**) with and without the addition of PAβN in IMP-6-producing *E. coli* strains.

**Table 1 antibiotics-11-00032-t001:** Antimicrobial susceptibility among 42 strains of IMP-6-producing *E. coli*.

	Susceptible	Intermediate	Resistant	MIC_50_ (μg/mL)	MIC_90_ (μg/mL)
IPM	21 (50.0%)	14 (33.3%)	7 (16.7%)	1.5	4
MEPM	1 (2.4%)	2 (4.8%)	39 (92.9%)	16	32
ETP	0 (0%)	3 (7.1%)	39 (92.9%)	16	64
DRPM	11 (26.2%)	4 (9.5%)	27 (64.3%)	8	32
PIPC	1 (2.4%)	0 (0%)	41 (97.6%)	>64	>64
CAZ	0 (0%)	1 (2.4%)	41 (97.6%)	>16	>16
CFPM	1 (2.4%)	1 (2.4%)	40 (95.2%)	>16	>16
CPFX	1 (2.4%)	0 (0%)	41 (97.6%)	>2	>2
LVFX	1 (2.4%)	0 (%)	41 (97.6%)	>4	>4
AMK	41 (97.6%)	1 (2.4%)	0 (0%)	<4	16
GM	11 (26.2%)	20 (47.6%)	11 (26.2%)	8	>8

**Table 2 antibiotics-11-00032-t002:** Characteristics of carbapenemase and ESBL among 42 strains of IMP-6-producing *E. coli*.

	Carbapenemase Producing	Carbapenemase Genes	ESBL Producing	ESBL Genes		
		*bla* _IMP-6_		*bla* _CTX-M-2_	*bla* _CTX-M-14_	*bla* _CTX-M-15_
Number of isolates (*n* = 42)	42	42	29	17	1	2
(%)	100.0	100.0	69.0	40.5	2.4	4.8

**Table 3 antibiotics-11-00032-t003:** Plasmid replicon typing among IMP-6-producing *E. coli* strains.

	Total (*n* = 42)	IMP-6 + CTX-M-2 (*n* = 17)	IMP-6 (*n* = 25)	*p*-Value
FIA	37 (88.1%)	15 (88.2%)	22 (88.0%)	1
FIB	28 (66.7%)	9 (52.9%)	19 (76.0%)	0.184
FIC	0 (0.0%)	0 (0.0%)	0 (0.0%)	N.A.
F	40 (95.2%)	17 (100.0%)	23 (92.0%)	0.506
FII	0 (0.0%)	0 (0.0%)	0 (0.0%)	N.A.
HI-1	0 (0.0%)	0 (0.0%)	0 (0.0%)	N.A.
HI-2	0 (0.0%)	0 (0.0%)	0 (0.0%)	N.A.
I1	18 (42.9%)	1 (5.9%)	17 (68.0%)	<0.001 *
L/M	0 (0.0%)	0 (0.0%)	0 (0.0%)	N.A.
N	36 (85.7%)	16 (94.1%)	20 (80.0)	0.374
P	0 (0.0%)	0 (0.0%)	0 (0.0%)	N.A.
W	0 (0.0%)	0 (0.0%)	0 (0.0%)	N.A.
T	0 (0.0%)	0 (0.0%)	0 (0.0%)	N.A.
A/C	2 (4.8%)	0 (0.0%)	2 (8.0)	0.506
K	0 (0.0%)	0 (0.0%)	0 (0.0%)	N.A.
B/O	4 (9.5%)	2 (11.8%)	2 (8.0)	1
X	0 (0.0%)	0 (0.0%)	0 (0.0%)	N.A.
Y	0 (0.0%)	0 (0.0%)	0 (0.0%)	N.A.

N.A.: not applicable; * Statistical significance.

## Data Availability

All data generated or analyzed during this study are included in this published article.

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
