# Peer review of "The Antimicrobial Resistance Characteristics of Imipenem-Non-Susceptible, Imipenemase-6-Producing Escherichia coli"

_antibiotics, 2021, doi:10.3390/antibiotics11010032_

Round 1

Reviewer 1 Report

Thank you for giving me the opportunity to review the manuscript "The epidemiological and antimicrobial resistance characteristics of imipenem-non-susceptible, imipenemase-6-producing Escherichia coli". The manuscript touches an important subject, imipenemase-6 (IMP-6) type carbapenemase-producing Enterobacteriaceae.  This carbapenemase  is dangerous due to its unique lack of antimicrobial susceptibility.  In addition to carbapenemase, outer membrane porins and efflux pumps also play roles in carbapenem resistance by reducing the antimicrobial concentration inside cells. The researchers found out in the 42 studied isolated  50.0%  IMP-6 producing strains  non-susceptible to IPM, which is different from the typical pattern generally incountered. 

I believe a little work into the introduction is nedded, also in the discission part supplemantary references may be added.

Altoghether I find this research very important.

Author Response

Reviewer 1

Thank you for giving me the opportunity to review the manuscript "The epidemiological and antimicrobial resistance characteristics of imipenem-non-susceptible, imipenemase-6-producing Escherichia coli". The manuscript touches an important subject, imipenemase-6 (IMP-6) type carbapenemase-producing Enterobacteriaceae.  This carbapenemase is dangerous due to its unique lack of antimicrobial susceptibility.  In addition to carbapenemase, outer membrane porins and efflux pumps also play roles in carbapenem resistance by reducing the antimicrobial concentration inside cells. The researchers found out in the 42 studied isolated 50.0% IMP-6 producing strains non-susceptible to IPM, which is different from the typical pattern generally incountered.

I believe a little work into the introduction is nedded, also in the discission part supplemantary references may be added.

Altoghether I find this research very important.

(Amendments)

Thank you for your review and comment. We are keen to take your advice in revising the introduction and discussion. Could you please provide concrete suggestions in this additional work?

We are thankful for your appreciation in our research.

Reviewer 2 Report

The original article made by Onishi et al. is well and clear written. The aim of this study was to assess antimicrobial susceptibility and the effects of the three carbapenem resistance mechanisms, ESBL production, and epidemiology in IMP-6 producing Escherichia coli. The Authors described the research methods in detail and discussed the results of this study. In my oppinion, this article can add important data to enrich the knowledge in this area. Only minor editorial corrections are needed.

- in line 72 should be [multidrug] instead of [multi-drug]

- please write [mL] instead of [ml]; both in main text and in Supplemental Table 1

- each description of a table and figure should end with a dot

- in Figs. 1 and 2 – please use minus sign instead of hyphen for negative values in figure

- in line 154 – please put a space between [<] and [0.001]

- in line 183 please remove the double space between [China] and [showed]

- please put a space between the temperature value and the unit [°C], e.g. in lines 273-275, 296-298

- in line 267 was probably done Shift + Enter; it should be removed.

Author Response

Reviewer 2

The original article made by Onishi et al. is well and clear written. The aim of this study was to assess antimicrobial susceptibility and the effects of the three carbapenem resistance mechanisms, ESBL production, and epidemiology in IMP-6 producing Escherichia coli. The Authors described the research methods in detail and discussed the results of this study. In my oppinion, this article can add important data to enrich the knowledge in this area. Only minor editorial corrections are needed.

- in line 72 should be [multidrug] instead of [multi-drug]

(Amendments)

Thank you for the comment. I corrected to [multidrug].

(Page 2, Line 70)

- please write [mL] instead of [ml]; both in main text and in Supplemental Table 1

(Amendments)

Thank you for the comment. I corrected to [mL] throughout the text and Supplemental Table 1.

(Page 2, Line 98, Page 3, Line 102, Page 8, Line 267, Table 1, Figure 3 and Supplemental Table 1)

- each description of a table and figure should end with a dot

(Amendments)

Thank you for the comment. I add a dot where end of description in all tables and figures.

- in Figs. 1 and 2 – please use minus sign instead of hyphen for negative values in figure

(Amendments)

Thank you for the comment. I revised to “minus sign” for negative values.

(Page3, Line 117-119, Figure 1 and Figure 2)

- in line 154 – please put a space between [<] and [0.001]

(Amendments)

Thank you for the comment. I added a space between [<] and [0.001].

(Page 1, Line 33-34, Page 6, Line 161, and Table 3)

- in line 183 please remove the double space between [China] and [showed]

(Amendments)

Thank you for the comment. I corrected the space between [China] and [showed].

(Page 7, Line 186)

- please put a space between the temperature value and the unit [°C], e.g. in lines 273-275, 296-298

(Amendments)

Thank you for the comment. I put a space between temperature value and the unit [°C].

(Page 8, Line 277-279 and Page 9, Line 302-304)

- in line 267 was probably done Shift + Enter; it should be removed.

(Amendments)

Thank you for the comment. I revised the part you pointed out.

(Page 8, Line 272)

Reviewer 3 Report

The manuscript is very well structured, each one of the sections presents the information in an adequate and detailed way, which allows a better understanding of each of the aspects dealt with. The supporting information, provided in the introduction and the discussion, is appropriate to the topics developed.
However, I consider that the title is not consistent with the main objectives of the work and its development. In this case, the word epidemiology is irrelevant, since at no time are epidemiological data on resistant bacteria, or the prevalence of some of these bacteria in different geographical areas, being developed in depth. In fact, the samples taken to obtain the corresponding cultures are not significant to ensure a geographic prevalence. Also, in the conclusion, the epidemiological situation is not described. For this reason, I suggest removing that word from the title, or where appropriate, addressing in a detailed way the epidemiology of the bacteria studied.

Author Response

Reviewer 3

The manuscript is very well structured, each one of the sections presents the information in an adequate and detailed way, which allows a better understanding of each of the aspects dealt with. The supporting information, provided in the introduction and the discussion, is appropriate to the topics developed.

However, I consider that the title is not consistent with the main objectives of the work and its development. In this case, the word epidemiology is irrelevant, since at no time are epidemiological data on resistant bacteria, or the prevalence of some of these bacteria in different geographical areas, being developed in depth. In fact, the samples taken to obtain the corresponding cultures are not significant to ensure a geographic prevalence. Also, in the conclusion, the epidemiological situation is not described. For this reason, I suggest removing that word from the title, or where appropriate, addressing in a detailed way the epidemiology of the bacteria studied.

(Amendments)

Thank you for the suggestion. We removed “epidemiology” from title as below.

The antimicrobial resistance characteristics of imipenem-non-susceptible, imipenemase-6-producing Escherichia coli

Reviewer 4 Report

The research concern the important problem of carbapenem resistance, and shows that new genetic variants of resistant strains are emerging at an alarming rate. The studies were carried out correctly, the Manuscript is clear and easy to read. It is important to continue the undertaken topic.

Author Response

Reviewer 4

The research concern the important problem of carbapenem resistance, and shows that new genetic variants of resistant strains are emerging at an alarming rate. The studies were carried out correctly, the Manuscript is clear and easy to read. It is important to continue the undertaken topic.

(Amendments)

Thank you for your review and appreciation.